# Learning Multi-Object Dynamics with Compositional Neural Radiance Fields

**Danny Driess**
TU Berlin

**Zhiao Huang**
UC San Diego

**Yunzhu Li**
MIT

**Russ Tedrake**
MIT

**Marc Toussaint**
TU Berlin

**Abstract:** We present a method to learn compositional multi-object dynamics models from image observations based on implicit object encoders, Neural Radiance Fields (NeRFs), and graph neural networks. NeRFs have become a popular choice for representing scenes due to their strong 3D prior. However, most NeRF approaches are trained on a single scene, representing the whole scene with a global model, making generalization to novel scenes, containing different numbers of objects, challenging. Instead, we present a compositional, object-centric auto-encoder framework that maps multiple views of the scene to a *set* of latent vectors representing each object separately. The latent vectors parameterize individual NeRFs from which the scene can be reconstructed. Based on those latent vectors, we train a graph neural network dynamics model in the latent space to achieve compositionality for dynamics prediction. A key feature of our approach is that the latent vectors are forced to encode 3D information through the NeRF decoder, which enables us to incorporate structural priors in learning the dynamics models, making long-term predictions more stable compared to several baselines. Simulated and real world experiments show that our method can model and learn the dynamics of compositional scenes including rigid and deformable objects. Video: https://dannydriess.github.io/compnerfdyn/

**Keywords:** Neural Radiance Fields, Dynamics Models, Graph Neural Networks

## 1   Introduction

Learning models from observations that predict the future state of a scene is a fundamental concept for enabling an agent to reason about actions to achieve a desired goal. A major challenge in learning predictive models is that raw observations such as images are usually high-dimensional. Therefore, a common approach is to map the observation space into a lower-dimensional latent representation of the scene via an auto-encoder structure. Based on those latent vectors, a dynamics model can be learned that predicts the next latent state, conditioned on actions an agent takes. An intuition for this is that if a latent vector is sufficient to reconstruct the observations, then it contains enough information about the scene to learn a dynamics model on top of it. While an auto-encoder structure combined with a latent dynamics model is a general approach that is applicable for a large variety of tasks, it raises multiple challenges. First, scenes in our world are *composed* of multiple objects. Therefore, a fixed-size latent vector has difficulties in generalizing over different and changing numbers of objects in the scene than during training, both due to the limited capacity of fixed-size vectors and lack of diversity in the training distribution. Second, image observations are 2D, but the 3D structure of our world is essential for many tasks to reason about the underlying physical processes governing the dynamics the model should predict. Dealing with occlusions, object permanence, and ambiguities in 2D views is challenging for 2D image representations. Importantly, many forward predictive models in visual observation spaces suffer from instabilities in making long-term predictions, often manifested in blurry image predictions [1].

One way to address these issues is to incorporate inductive biases and structural priors in the model architectures. Li et al. [2] proposed to use Neural Radiance Fields (NeRFs) [3] as a decoder within an auto-encoder to learn dynamics models in latent spaces. NeRFs exhibit strong structural priors about the 3D world, leading to increased performance over 2D baselines. However, the approach of [2] represents the whole scene as a single latent vector, which we found insufficient for scenes composed of multiple, different numbers of objects, in terms of representation and dynamics prediction.

In the present work, we aim to overcome these challenges by incorporating inductive biases on the compositional nature and underlying 3D structure of our world both in learning the latent representa-

6th Conference on Robot Learning (CoRL 2022), Auckland, New Zealand.

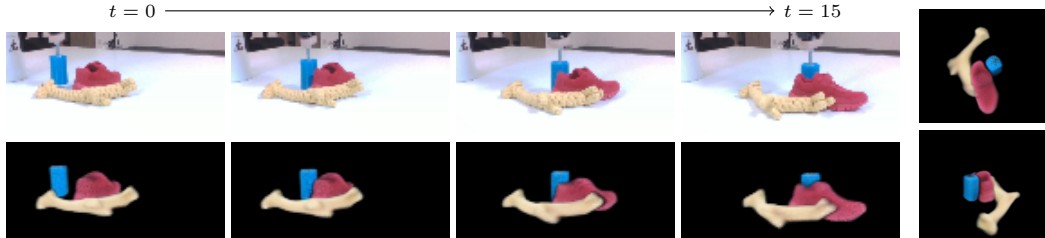

(a) Bottom row renderings of forward predictions with dynamic model, top row ground truth   (b) novel view

Figure 1: Visual forward predictions with our model. (a) left: initial scene, (a) right: after 15 prediction steps into the future. (b) renderings of the model after 15 prediction steps from novel views. Future predictions are based on the initial observation of the scene at $t = 0$ and then rendered from the prediction of the latent vectors. Blue pusher is articulated by the robot. Despite multiple objects interacting, predictions are sharp and accurate.

tions themselves and the dynamics model. We propose a compositional, object-centric auto-encoder framework whose latent vectors are used to learn a compositional forward dynamics model in that learned latent space based on graph neural networks (GNN). More specifically, we learn an implicit object encoder that maps image observations of the scene from multiple views to a set of latent vectors that each represent an object in the scene separately. These latent object encodings then parameterize individual NeRFs for each object. We apply compositional rendering techniques to synthesize images from multiple viewpoints, which forces the object-centric NeRF functions and the corresponding latent vectors to learn precise 3D configurations of the constituting objects. This 3D inductive bias both in the encoder and the compositional NeRF decoder enables us to incorporate priors from the models' own predictions about objects interactions via an estimated adjacency matrix into learning the GNN dynamics model, making long-term dynamics predictions more stable. This long term-stability allows us utilize a planning method based on RRTs in the latent space.

In our evaluations, we show through comparisons that non-compositional auto-encoder frameworks and non-compositional dynamics models struggle with tasks containing multiple objects, while our framework generalizes well over different numbers of objects than during training and is capable of generating sharp and stable long-term predictions. Relative to more traditional multibody system identification [4], these models learn the geometry of unknown objects in addition to (implicitly) learning the inertial and contact parameters. We demonstrate the performance of the approach in terms of image reconstruction error, dynamics prediction error, and planning, generalizing over different numbers of objects than during training. Our experiments include rigid and deformable objects in simulation and with a real robot. To summarize, our main contributions are

- A compositional scene encoding framework that uses implicit object encoders and NeRF decoders for each object, forcing the view-invariant latent representation to learn about the 3D structure of the problem in a composable way.
- A factored dynamics model in the latent space as a graph neural network (GNN), exploiting the compositional nature of the scene representation and an adaptive adjacency matrix estimated from the model itself to yield stable long-term predictions.

## 2   Related Work

**Learning Dynamics Models for Compositional Systems.** Graph neural networks (GNNs) have shown success in introducing relational inductive biases [5], enabling them to model the dynamics of compositional systems consisting of interactions between multiple objects [6, 7, 8, 9, 10, 11], large-scale dynamical systems represented with particles and meshes [12, 13, 14, 15, 16, 17], or from visual observations [18, 19, 20, 21, 22, 23, 24]. Our method differs from prior work by learning compositional scene representations grounded in 3D space from visual observations. Our novel combination of implicit object encoders and graph-based neural dynamics models reflects the structure of the underlying scene, which endows our agent with better generalization ability in handling complicated compositional dynamic environments.

**NeRF for Compositional and Dynamic Scenes.** Recent advances on neural implicit representations [25] have demonstrated widespread success in image synthesis or 3D reconstruction [26, 27, 28, 29, 30]. Notably, Neural Radiance Fields (NeRF) show impressive results on novel-view synthesis [3]. Initial NeRF approaches were trained on a single scene without generalization. Prior work [31, 32, 33, 34, 35, 36] have since proposed to modify neural scene representations to make them compositional for static scenes without considering dynamics of object interactions. Re-

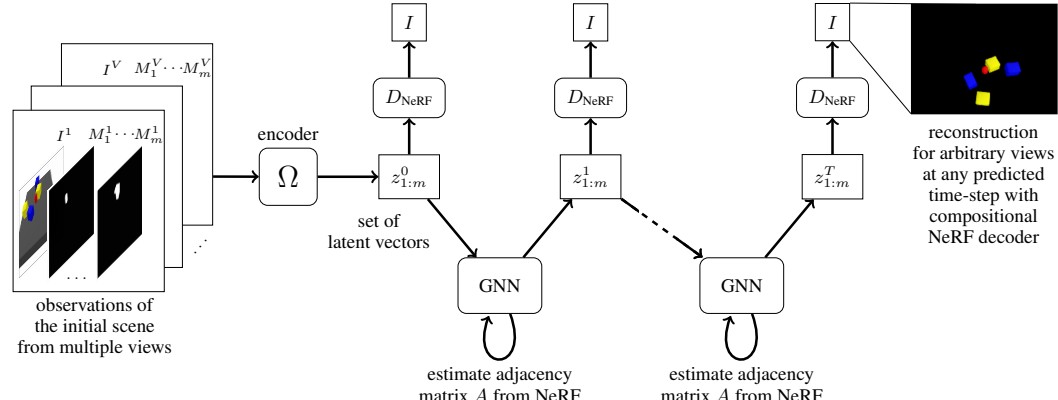

Figure 2: Overview of the dynamics prediction framework. The initial scene observations are encoded with $\Omega$ into a set of latent vectors $z_{1:m}$, each representing the objects individually. The GNN dynamics model predicts the evolution of the latent vectors. At each step, the predicted latent vectors can be rendered into an arbitrary view with the compositional NeRF decoder. Refer to the appendix for visualizations of $\Omega$ and the GNN.

cent research has also extended NeRF to enable view synthesis from a sparse set of views [37], as well as modeling dynamic scenes by learning implicitly represented flow fields or time-variant latent codes [38, 39, 40, 41, 42, 43, 44, 45, 46, 33, 47]. However, these approaches for dynamic environments typically interpolate over a single time sequence and are not able to handle scenes of different initial configurations or different action sequences, limiting their use in downstream planning and control tasks. Li et al. [2] addressed this issue by combining an NeRF auto-encoding framework with modeling the dynamics in a latent space. Yet, they employed a single latent vector as the whole scene representation, which we will show is insufficient at modeling compositional systems. In contrast, our method considers a graph-based scene representation to capture the structure of the underlying scene and achieves significantly better generalization performance than [2].

**Implicit Models in Robotics.** Implicit models in robotics have been explored, e.g., for grasping [48, 49, 50, 51], deformable objects [52], object descriptors [53], or general manipulation constraints [54, 55]. Analytic signed distance functions (SDFs) [56, 57, 54] or learned NeRFs [58] are used for trajectory planning. One assumption in [54, 55] is that SDF values are available during training. Our work, in contrast, directly operates on RGB images without requiring explicit 3D shape supervision.

## 3 Overview – Compositional Visual Dynamics Learning

Our dynamics learning framework (Fig. 1) consists of three parts, an object encoder $\Omega$ turning observations into a set of latent vectors $z_{1:m}$, a compositional NeRF-based decoder $D_{\text{NeRF}}$ that renders the latent vectors back into images of the scene to train the encoder, and a graph neural network dynamics model $F_{\text{GNN}}$ predicting the evolution of the scene in the latent space. This section gives a high-level overview, while Sec. 4, Sec. 5 as well as the appendix Sec. C, Sec. D provide details.

Assume that a scene is observed by RGB images $I^i \in \mathbb{R}^{3 \times h_I \times w_I}$, $i = 1, \ldots, V$ from $V$ many camera views and that the scene contains $m$ objects $j = 1, \ldots, m$. We further assume to have access to the camera projection matrices $K^i \in \mathbb{R}^{3 \times 4}$ for each view and binary masks $M_j^i \in \{0, 1\}^{h_I \times w_I}$ of each object $j$ in view $i$. Given those posed images and masks, the goal is to learn an encoder $\Omega$ that fuses the information of the objects observed from the multiple views into a set of latent vectors $z_{1:m}$ by querying $\Omega$ on the individual masks $M_j^{1:V}$ such that

$$z_j = \Omega \left( I^{1:V}, K^{1:V}, M_j^{1:V} \right) \in \mathbb{R}^k \tag{1}$$

represents the object $j$ separately. $\Omega$ is trained end-to-end with a NeRF decoder $D_{\text{NeRF}}$ reconstructing

$$I = D_{\text{NeRF}}(z_{1:m}, K) \tag{2}$$

for arbitrary views specified by the camera matrix $K$ from the set of latent object representations $z_{1:m}$. The initial observation of the scene is encoded with $\Omega$ into the initial latent vectors $z_{1:m}^0$. The GNN dynamics model $z_{1:m}^{t+1} = F_{\text{GNN}}(z_{1:m}^t)$ then generates long-term predictions of future latent states $z_{1:m}^t$ that can also be decoded with $D_{\text{NeRF}}$ to yield visual predictions from arbitrary views.

# 4 Encoding Scenes with Compositional Image-Conditioned NeRFs

## 4.1 Implicit Object Encoder

Instead of learning $\Omega$ defined in (1) as a direct mapping from images, camera matrices and masks to the latent vectors, we first encode each object in the scene as a feature-valued *function* over 3D space, conditioned on the image observations. This allows us to incorporate multiple views of the objects in a geometrically consistent way, as well as to apply 3D affine transformations to the objects, which will be important for the dynamics model (Sec. 5). This function is then turned into a latent vector by evaluating it on a workspace set followed by a 3D convolutional network.

All object feature functions are based on the *same* feature encoder $E(I^i, K^i(x)) \in \mathbb{R}^{n_o}$ that outputs an $n_o$-dimensional feature vector from the image $I^i$ of view $i$ at any 3D world coordinate $x \in \mathbb{R}^3$. This is realized by first projecting $x$ into camera coordinates $K^i(x) = \left(u^i(x), v^i(x), d^i(x)\right)^T \in \mathbb{R}^3$ where $u^i(x), v^i(x)$ are pixel coordinates in the image plane and $d^i(x) \in \mathbb{R}$ is the depth of $x$ from the camera origin. Hence, $E$ is a function of the camera coordinates only and not of absolute world coordinates. Using bilinear interpolation, the encoder $E(I^i, K^i(x))$ determines the RGB values of $I^i$ at $(u^i(x), v^i(x))$ which are passed through a dense neural network (MLP). Parallel to this, a dense MLP encoding of $K^i(x)$ is computed. The concatenated outputs of both MLPs define the encoding feature vector $E(I^i, K^i(x))$. Intuitively, $E(I^i, K^i(x))$ is a feature vector computed from what can be seen of the world at $x$ in the image $I^i$ from viewpoint $i$, taking into account its location relative to the camera origin of the view $i$, which is important not only to enable the model to reason about the 3D geometry, but also to enable us to obtain a functional representation of a specific object $j$. Namely, we define the feature function for object $j$ by summing over the individual views $i$

$$y_j(x) = \frac{1}{p(x)} \sum_{i:\ K^i(x) \in M_j^i} E(I^i, K^i(x)) \in \mathbb{R}^{n_o} \quad \text{with} \quad p(x) = \sum_{i:\ K^i(x) \in M_j^i} 1. \qquad (3)$$

Importantly, for a specific $x$, this sum only takes those views $i$ into account where the object $j$ can be seen, i.e., where the camera coordinates $K^i(x)$ of $x$ are within the object's mask $M_j^i$. We define $y_j(x) = 0 \in \mathbb{R}^{n_o}$ if $p(x) = 0$, meaning if an object is not observed from any view at $x$, the corresponding feature vector is zero. An advantage of this formulation is that it naturally handles occlusions in different views and fuses the observations from different views consistently.

Given the implicit object descriptor function $y_j(\cdot)$ of object $j$, we turn it into a latent vector $z_j \in \mathbb{R}^k$ representing object $j$ with a 3D convolutional network $\Phi$ as follows. Formally, $z_j = \Phi(y_j)$ is a function of the object *function*. As discussed in [54], learning a function of a function can be realized with neural networks by evaluating $y_j$ on a workspace set. We assume that the interactions in the scene happen within a workspace set $\mathcal{X} \subset \mathbb{R}^3$ that is large enough to contain all objects. This workspace set is discretized as the voxel grid $\mathcal{X}_h \in \mathbb{R}^{d \times h \times w}$. The object descriptor functions are then evaluated on $\mathcal{X}_h$ which produces an object feature voxel grid that is processed with a 3D convolutional neural network leading to the latent vector $z_j \in \mathbb{R}^k$, i.e.

$$z_j = \Phi(y_j) = \text{CNN}(y_j(\mathcal{X}_h)). \qquad (4)$$

Note that the same workspace set $\mathcal{X}_h$ is used for all objects. The appendix (Sec. C) contains visualizations of the architectures of $E$, $y$ and $\Phi$ (Fig. 8, 9, 10).

In summary, the object encoder $z_{1:m} = \Omega\left(I^{1:V}, K^{1:V}, M_{1:m}^{1:V}, \mathcal{X}_h\right)$ maps images from multiple views, object masks and the set $\mathcal{X}_h$ to latent vectors. The resulting $z_j$'s contain not only the appearance of the objects, but also their spatial configurations in the scene relative to other objects.

## 4.2 Decoder as Compositional, Conditional NeRF Model

The general idea of NeRF [3] is to learn a function $f$ that predicts at a 3D world coordinate $x \in \mathbb{R}^3$ the RGB color value $c(x) \in \mathbb{R}^3$ and volume density $\sigma(x) \in \mathbb{R}_{\geq 0}$. Based on $(\sigma(\cdot), c(\cdot)) = f(\cdot)$, images from arbitrary views and camera configurations can be rendered by determining the color of the pixels along corresponding camera rays through volumetric rendering. For details, see Sec. C.

Compared to this standard NeRF formulation where one single model is used to represent the whole scene, we associate separate NeRFs with each object, meaning that the NeRF for object $j$

$$(\sigma_j(x), c_j(x)) = f_j(x) = f(x, z_j) \qquad (5)$$

is conditioned on $z_j$ for $j = 1, \ldots, m$. $\sigma_j$ is the density and $c_j$ the color prediction for object $j$, respectively. To turn those $f_{1:m}$ back into a global NeRF model that can be rendered to an image, we sum the individual predicted object densities $\sigma(x) = \sum_{j=1}^{m} \sigma_j(x)$ and obtain the colors as their density weighted combination $c(x) = \frac{1}{\sigma(x)} \sum_{j=1}^{m} \sigma_j(x) c_j(x)$. These composition formulas have been proposed multiple times in the literature, e.g. [59, 32]. This composition forces the individual NeRFs to learn the 3D configuration of each object individually and therefore ensures that each $f_j$ only predicts the object where it is located in the 3D space.

To summarize, the compositional NeRF-decoder $D_{\text{NeRF}}$ takes the set of latent vectors $z_{1:m}$ for objects $j = 1, \ldots, m$ and the camera matrix $K$ for a desired view as input to render $I = D_{\text{NeRF}}(z_{1:m}, K)$. Since we only represent the objects and not the background as NeRFs, rendering the composed NeRF will yield an image with the background subtracted. In the experiments, we investigate the importance of the decoder being both compositional and a NeRF.

### 4.3 Training

The auto-encoder framework is trained end-to-end on an $L_2$ image reconstruction loss for view $i$

$$\mathcal{L}^i = \sum_{(u,v) \in \hat{M}_{\text{tot}}^i} \left\| \left( I^i \circ M_{\text{tot}}^i \right)_{uv} - D_{\text{NeRF}} \left( \Omega \left( I^{1:V}, K^{1:V}, M_{1:m}^{1:V}, \mathcal{X}_h \right), K^i \right)_{uv} \right\|_2^2. \tag{6}$$

Since solely the objects are represented as NeRFs and not the background, we compute the union of the masks of the individual objects $M_{\text{tot}}^i = \bigvee_{j=1}^{m} M_j^i$ and define the target image as $I^i \circ \hat{M}_{\text{tot}}^i$ with $\hat{M}_{\text{tot}}^i$ being a slightly enlarged union mask. Please refer to the appendix Sec. C for more details.

## 5   Latent Dynamics Model with Graph Neural Networks

Having trained the auto-encoder framework, we learn a graph neural network dynamics model

$$z_{1:m}^{t+1} = F_{\text{GNN}} \left( z_{1:m}^t, A^t \right) \tag{7}$$

in the latent space, where $A^t \in \{0, 1\}^{m \times m}$ is the adjacency matrix at time $t$. Following [8], we use multi-step message passing to deal with cases where multiple objects interact within one prediction step. Refer to the appendix Sec. D and Algo. 1 for more details about our GNN dynamics model.

**Adjacency Matrix from Learned Model.** The adjacency matrix $A$ in the GNN dynamics model (7) plays an important role in indicating which objects interact. While a dense adjacency matrix, i.e. a graph where each object interacts with all other objects, would in principle work as the GNN could figure out from the latent vectors which objects interact, we found that the long-horizon prediction performance is greatly increased if $A$ is more selective in reflecting which objects actually interact.

We propose to utilize the NeRF decoder density prediction $\sigma_j$ for each object to determine the adjacency matrix from the models' own predictions during training and planning. In order to do so, we define the entries of the adjacency matrix between objects $i$ and $j$ based on the collision integral

$$A_{ij} = \begin{cases} 1 & \int_{\mathcal{X}} [\sigma(x, z_i) > \kappa][\sigma(x, z_j) > \kappa] \, \mathrm{d}x > 0 \\ 0 & \text{else} \end{cases} \tag{8}$$

over the density predictions of the learned NeRF model for a threshold $\kappa \geq 0$. Estimating $A$ this way takes the actual 3D geometry of the objects in the scene into account and thereby informs the GNN dynamics model, leading to more stable predictions. Please refer to the appendix Sec. D for more details about $A$ and how it is used in the forward prediction Algo. 1.

**Actions.** So far, we have formulated the GNN dynamics model without a notion of actions. We interpret an action as an intervention to a latent vector and train the GNN to predict the latent vectors at the next time step as a result to this modification. This allows us to not explicitly distinguish between controlled and uncontrolled/passive objects. In order to realize these interventions and hence to incorporate actions in the first place, we utilize the fact that our object encoder is built from an implicit representation. Assume that an action is a rigid transformation $q \in \mathbb{R}^7$ applied on object $j$. As described in Sec. C we can modify the object's latent vector $z_j^t$ into the transformed $\bar{z}_j^t = z_j^{t+1} = \mathcal{T}(q)[z_j]$ representing the rigidly transformed object $j$. The model $F_{\text{GNN}}$ then predicts how the other objects in the scene react to this rigid transformation of the articulated object.

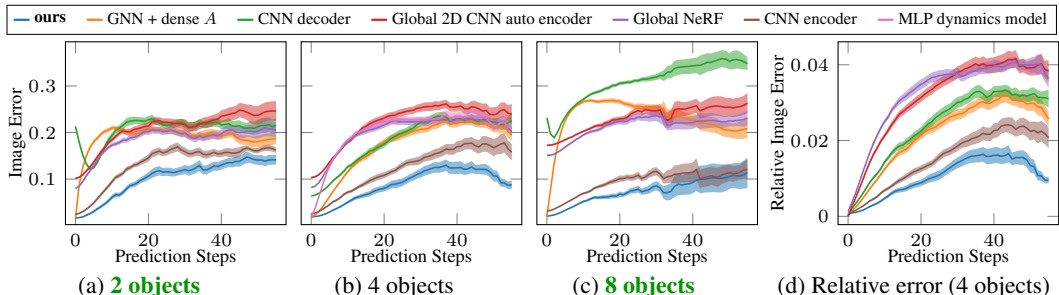

|  |  |  |  |
|---|---|---|---|
| (a) **2 objects** | (b) 4 objects | (c) **8 objects** | (d) Relative error (4 objects) |

Figure 3: Image prediction error comparison between the reconstructed image from the predicted latent vectors over the number of time steps into the future and ground truth image observations, for test dataset of scenarios containing 2, 4, and 8 objects (plus the pusher). 2 and 8 objects is generalization over number of objects, 4 is as during training. One step corresponds to 2 cm movement, i.e. for 50 steps the pusher has moved 1 m.

## 6 Experiments

We demonstrate our framework on pushing tasks both in simulation and in the real world. The scenarios are challenging as they are composed of multiple, interacting objects, sparse rewards, and complex dynamics [60, 61, 62, 63]. Please refer to the video `https://dannydriess.github.io/compnerfdyn/` as well as the appendix for more details and further experiments.

### 6.1 Visual Reconstruction and Prediction Performance – Comparison to Baselines

We compare our framework to non-compositional scene representations, non-compositional dynamics models, 2D CNN baselines (visual foresight) without NeRF as decoder, and the importance of estimating the adjacency matrix from the model itself.

**Reconstruction and Prediction Performance for Generalization over Numbers of Objects.** Fig. 4a shows predictions of the model forward unrolled in time for an action sequence of the red pusher, i.e. applying Algo. 1 (appendix) to an initial scene observation and rendering the predicted latent vectors with the NeRF decoder. Despite the movements in this scene leading to multiple object interactions, even after 38 time steps, the rendered predictions from the model are still sharp and reflect the underlying dynamics. By utilizing the estimated adjacency matrix, there is little drift in the objects, leading to long-term prediction stability. Due to its compositional nature, our model generalizes to scenes that contain more or less objects than in the training set, as shown in Fig. 5 where eight objects plus the pusher are observed and reconstructed with high quality from novel views, although during training the model has seen only and exactly 4 objects.

**Comparison to Non-Compositional Scene Representation Baselines.** We compare to two non-compositional baselines where the scene is represented globally with one single latent vector per time-step. The dynamics model for these baselines is an MLP $z^{t+1} = F_{\text{MLP}}(z^t, q)$ that takes the action $q$ as an additional input. The first baseline (Global NeRF) is the approach from [2], i.e. we use their CNN encoder to produce one latent vector that conditions a global NeRF which reconstructs the whole scene (Fig. 14c). The second baseline (Global 2D CNN auto encoder) uses both a 2D CNN encoder and 2D CNN decoder as well as a single latent vector representing the whole scene (Fig. 4c). Such frameworks have been used many times in the literature, e.g. [1, 64, 65, 66, 67] and are known as visual foresight. Fig. 3 shows that both global baselines are significantly inferior in our scenarios to our proposed compositional framework, especially for long horizons.

**Comparison to 2D Baselines – Importance of NeRF as Decoder.** In this section, we replace the NeRF decoder with a 2D CNN decoder to investigate the importance of NeRFs. This decoder takes as input one single latent vector and the camera matrix, i.e. $I = D_{\text{CNN}}(z, K)$. In order to make it compositional, we aggregate the set of latent vectors $z_{1:m}$ from $\Omega$ with a mean operation and then pass the aggregated feature through an MLP to produce the single $z$ for $D_{\text{CNN}}$. The rest of the architecture, i.e. implicit object encoder and GNN, stays the same. Since there is no clear way to estimate the adjacency matrix from $D_{\text{CNN}}$, we use a dense adjacency matrix for the GNN. As one can see in Fig. 3, the long-term prediction performance of the CNN decoder is significantly worse than with a compositional NeRF model as the decoder, especially when asking for numbers of objects that differ from the training distribution. Qualitatively, one can see in Fig. 4c that not only the initial reconstruction is much less sharp compared to the NeRF-based models, but especially also that even after only a few time-steps, the predictions with the CNN decoder are of little use.

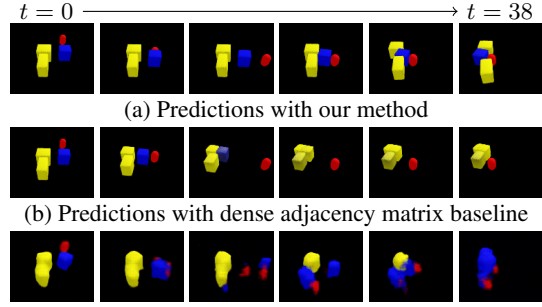

$t = 0$ ⟶ $t = 38$

(a) Predictions with our method

(b) Predictions with dense adjacency matrix baseline

(c) Predictions with CNN decoder baseline (no NeRF)

Figure 4: Visual forward predictions. With our proposed method (a), the predictions are very sharp, even after 38 steps, while with a dense adjacency matrix (b) leads to drifting objects until the predictions are not useful anymore. The CNN decoder baseline is even worse, such that after only a few steps the predictions are of little use. Multiple object interactions happen in this scene.

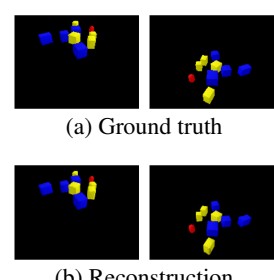

(a) Ground truth

(b) Reconstruction

Figure 5: Generalization to twice as many objects as during training.

**Comparison to CNN Encoder.** Exchanging the implicit object encoder with a 2D CNN compositional encoder leads to an auto-encoder framework similar to [32]. As seen in Fig. 3, the performance is better compared to the other baselines, but still clearly worse than with the proposed method.

**Importance of Estimating the Adjacency Matrix.** In Sec. 5, we propose how the adjacency matrix of the GNN can be estimated from the learned NeRFs to increase the long-term stability of the predictions. Here we compare to a dense adjacency matrix, i.e. where the network has to figure out from the latent vectors themselves which objects interact. As one can see in Fig. 3 and Fig. 4b, a dense $A$ has significantly worse long-horizon prediction performance compared to our proposed way of estimating $A$ through the learned NeRF model. In the 2 and 8 object case (generalization over numbers of objects), the predictions with the dense $A$ are useless after only a few time-steps.

**Non-Compositional Dynamics.** Replacing the GNN with a fully connected MLP $z_{1:m}^{t+1} = F_{\text{MLP}}(z_{1:m}^t)$ leads to worse performance than with a GNN with dense adjacency matrix. This model cannot generalize to different numbers of objects due to its fixed input size.

**Summary of Performance Comparisons** Our method outperforms all baselines both in terms of pure reconstruction error (as can be seen in Fig. 3 by the error after 0 prediction steps) *and* its ability to perform long-term predictions forward unrolled on the model's own predictions. Estimating the adjacency matrix from the model itself is important for long-term stability as it prevents objects from drifting away. Too large drift makes future predictions for a pushing tasks meaningless. Since the reconstruction error of our proposed method without dynamics is better than the baselines, the question arises if the increased performance is an artifact of the lower reconstruction error. We show in Fig. 3d the error in the image space between renderings when having access to the observations at each step and the renderings from the predicted latent vectors into the future after observing the scene only at the beginning. This shows the increase in error relative to the reconstruction process. The results indicate that not solely the reconstruction itself is the reason for the better performance, but that the structural choices of our framework also enable to learn the dynamics more precisely.

## 6.2 Planning and Execution Results on Object Sorting Task

To demonstrate the effectiveness of the learned model, we utilize it to solve a box sorting task, where the pusher needs to push colored boxes into their corresponding goal regions as shown in Fig. 6. This task is inspired by [68] and involves multiple challenges: As multiple objects interact, a greedy strategy of pushing objects straight to the goal region fails. Movements, i.e. actions, of the pusher do often not immediately lead to a change in the cost function, since contact with the object from a suitable side has to be established [54, 63]. In the appendix Sec. E we propose a latent space RRT that uses our framework for planning. Refer to the appendix and the video for more details about our proposed planning algorithm and comparisons to baselines.

## 6.3 Real World Experiments & Deformable Objects

Fig. 1a shows the rendered forward predictions of our model for a real world scenario where a robot pushes a shoe and a giraffe-shaped toy. Fig. 1b are renderings from novel view points. We further show in Fig. 7 that our method is also applicable to deformable objects. The appendix Sec. G contains more details regarding this experiment, including a quantitative analysis.

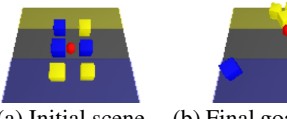 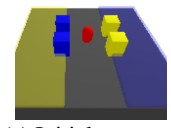 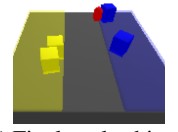

| (a) Initial scene | (b) Final goal achieved | (c) Initial scene | (d) Final goal achieved |

Figure 6: Two planning scenarios with our learned visual dynamics model and latent space RRT. The goal is to move the blue and yellow boxes into their respective shaded areas. (a), (c) are initial states, (b), (d) shows the achieved goal at the end of the planning/execution loop.

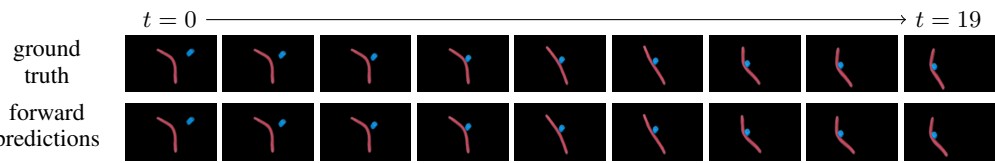

Figure 7: Forward predictions for deformable object scenario. The predicted reconstructions are based on the latent vector from the initial observation ($t = 0$) that is then forward predicted with the dynamics model.

## 7 Discussion & Limitations

The appendix Sec. A contains an extended discussion regarding limitations of this work.

**Object Masks.** The compositional scene encoding framework requires object masks to achieve compositionality. Many mature methods for instance segmentation have been developed such that we believe having masks as input is a reasonable assumption and we have also shown the applicability of our method in real world scenarios where no precise masks are available. We further investigate in Sec. F.4 and Sec. G.2 the robustness of our method with respect to mask perturbations. Methods for unsupervised discovery of objects [32, 69] could also be integrated into our pipeline.

**Latent Representations.** We have shown the great benefits of a compositional latent representation as it not only provides generalization over different numbers of objects in the scene, but also leads to increased reconstruction and dynamics prediction performance compared to non-compositional baselines. Furthermore, latent representations compress observations, enabling efficient dynamics prediction. As each object in the scene is represented as a latent vector of finite size, one could argue that latent models are capable of mainly representing objects with shapes similar to the training distribution. We currently use a single neural network to represent all objects. Therefore, all variations in object appearance are controlled by this latent vector (of size 64 in our experiments). Despite this being a rather small latent space, our framework already exhibits interpolation capabilities for object shapes and poses. The variety of the scenes considered in this work, including the deformable object, can sufficiently be represented with this latent vector. Therefore, we are confident that by exposing the method to a more diverse set of objects and potentially increasing the size of the latent space, one could see generalization capabilities beyond the training object category distribution, which is an interesting area for future research.

**Long-Term Prediction Stability.** Our dynamics model framework exhibits significantly better long-term prediction stability compared to baselines. Our experiments indicate that this is due the structural biases enabled through (compositional) NeRFs. This stability allowed us to use the model for planning scenarios requiring long-horizons, which none of the baseline methods could support. However, we believe that there is still room for improvement regarding the prediction stability.

## 8 Conclusion

Visual dynamics models are of high interest to the computer vision and robotics community, as they avoid explicit shape model assumptions and imply end-to-end perception. However, to support manipulation planning and reasoning we need models that generalize strongly over objects and provide stable long-term predictions. In this paper we proposed a system that introduces 3D structural and compositional priors at various levels, namely compositional NeRFs, 3D implicit object encoders, and GNNs dynamics with an adaptive adjacency matrix. Together our system exhibits significantly stronger long-term prediction performance compared to multiple baselines without these priors or without compositionality, and supports using a latent space RRT planner. We have shown generalization over different numbers of objects, notably up to two times more than during training.

**Acknowledgments**

This research has been supported by the Deutsche Forschungsgemeinschaft (DFG, German Research Foundation) under Germany's Excellence Strategy – EXC 2002/1 "Science of Intelligence" – project number 390523135 and Amazon.com Services, LLC PO# #2D-06310236. Danny Driess thanks the International Max-Planck Research School for Intelligent Systems (IMPRS-IS) for the support. The authors thank Valentin Hartmann for discussions regarding RRTs.

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
