# OpenReview forum: "Learning Multi-Object Dynamics with Compositional Neural Radiance Fields"
_robot-learning.org/CoRL/2022/Conference — CoRL 2022 Poster_

### Official Review · Reviewer_eGyX · 2022-07-28

**Originality:** Good
**Technical Quality:** Very Good
**Clarity Of Presentation:** Good
**Impact:** 3

**Recommendation:**

Weak Reject: I recommend rejecting the paper, but will not argue for my recommendation if the majority of other reviewers have a different opinion.

**Summary:**

The paper leverages NeRF for multi-object scene representation from image observation in a compositional manner. NeRF based autoencoders are deployed to encode each object into latent vectors, which were used compositionally to learn a compositional forward dynamic model using GNN.  The learnt model generalized across objects and provides stable long-term prediction that helps with manipulation planning and reasoning. RRT based methods can then be used to plan in the latent space for downstream manipulation tasks.


**Issues:**

See above

**Quality Of The Limitations Section:**

Additional details required

**Reviewer Expertise:**

4: The reviewer is confident but not absolutely certain that the evaluation is correct

**Robotics Focus:**

Sufficient demonstration on hardware

**Strengths And Weaknesses:**

Strength:
- The paper shows that compositional NeRF and GNN can produce a latent space where long horizon prediction is stable. Examples are shown for both simulation and real environment. And thanks to the nature of NeRF, the latent space is agnostic to view points.
- The method can be generalized to the number of objects and handles the interaction between objects well, even when the object is deformable.
- Comprehensive ablation studies are performed to support the design choices of the model components.
Weaknesses:
- For experiment 6.1, using image error only to measure the long horizon prediction error is not so convincing. NeRF is known to be a much better decoder to represent static 3d scenes compared to CNN. Even without forward dynamic prediction, CNN is not expected to have better reconstruction quality. But it doesn’t necessarily mean the latent space learnt with a CNN decoder is not suitable for long horizon dynamic prediction. From the supplementary video, CNN decoder can also capture the pose of the objects reasonably well. It is worth reporting the pose error of objects as well. Also there are a few questions regarding figure 3.
  - MLP dynamic model is shown on fig 3(b) only.
  - Why does the prediction error go down after 40 steps?
  - The y-axis scale of fig 3c is different from the rest.
  - It would be great to have more visualizations when using Global NeRF as decoder in Fig 4. It is interesting to see how the compositionality improves the decoder.
- For experiment 6.2, only qualitative results are shown. More statistics would be helpful to determining the contribution of each module compared to the baseline. For example success rate, avg. time/step taken to complete the task, etc. Without them, the experiment is not robust enough to support the claim that the method produces better latent space for planning and reasoning for robotic tasks.
- Similarly, only qualitative results are shown for the real world experiments. According to the video, the image prediction is no longer sharp when the objects are textured. This makes the visual reconstruction quality questionable as the proposed method might not be able to scale to more complex objects (in terms of shape and textures).


**Summary Of Recommendation:**

The paper shows that compositional NeRF and GNN can produce a latent space where long horizon prediction is stable. But the supporting experiments are weak. See above for more details.

---

> ### Author Response · Authors · 2022-08-26
> **Thank you for your review!**
>
> **Comment:**
>
> Thank you very much for your detailed and constructive review!
>
> > For experiment 6.1, using image error only to measure the long horizon prediction error is not so convincing. NeRF is known to be a much better decoder to represent static 3d scenes compared to CNN. Even without forward dynamic prediction, CNN is not expected to have better reconstruction quality. But it doesn’t necessarily mean the latent space learnt with a CNN decoder is not suitable for long horizon dynamic prediction.
>
> Thank you, we totally agree that this is a relevant and important point, which we already tried to investigate in the original submission, as mentioned in Sec 6.1 under the “Summary of Performance Comparisons” paragraph and shown in Fig. 3d (and Figures 18b, 19b, 20b, 21b, 22b, 25b in the appendix).
>
> These figures show the prediction error relative to the methods’ own reconstructions when having access to the ground truth scene state at every time step in comparison to the forward predicted reconstruction from the initial scene only. This way, we can distinguish the error introduced by the reconstruction capabilities from the long-term prediction stability of the dynamics model to show that not solely the reconstruction itself is the reason for the better performance, but that by NeRF-supervision a latent space is learned that enables the method to learn the dynamics more precisely.
>
> > From the supplementary video, CNN decoder can also capture the pose of the objects reasonably well. It is worth reporting the pose error of objects as well.
>
> Since the method does not predict object poses, reporting a pose error would require a pose estimation method from the images, which is non-trivial in general, but especially the case for the blurry images predicted by the CNN. It would be unclear how much of the reported error would be introduced by the pose estimation method. In case a NeRF decoder is used, we can report the error of the objects’ center-of-masses, because from the NeRF predictions we can compute the predicted center-of-masses and compare them with the ground truth. Results are shown in Fig 14.
>
> > MLP dynamic model is shown on fig 3(b) only.
>
> An MLP dynamics model cannot generalize over different numbers of objects in the scene, as its input size is fixed. Fig 3a and Fig 3c show generalization to different numbers of objects (2 and 8) for models that have been trained on scenes containing 4 objects. Therefore, one cannot use the MLP dynamics model for these generalization experiments, which is the reason that it is only shown in Fig 3b. We have added a sentence in the paper to clarify this. Thank you.
>
> > Why does the prediction error go down after 40 steps?
>
> For those scenes where predictions after 40 steps are still possible (we terminate if an object is being pushed from the table), it turns out that objects that are being pushed multiple times by the random pusher movements are often pushed in a way that they again align more closely with the open-loop forward predictions. This behavior can also be seen in the video. We have added a small discussion in the paper (caption Fig. 14) explaining this as well, thank you for pointing this out.
>
> > The y-axis scale of fig 3c is different from the rest.
>
> Thank you, this is fixed in the revised version.
>
> > It would be great to have more visualizations when using Global NeRF as decoder in Fig 4. It is interesting to see how the compositionality improves the decoder.
>
> Thanks for pointing this out. We have added those visualizations to the paper (Fig. 4c) as well as a video (attached). Please note that this experiment is challenging for the Global NeRF baseline, as it was trained on scenes containing 4 objects but is now tested on scenes containing only 3 objects being pushed. The fact that the global baseline has many difficulties generalizing to different numbers of objects as in this example was one of the main motivations for this work.
>
> **Zip File:**
>
> /attachment/0a182a08f4f9aff481b22f32315eb5cf4861eeff.zip

---

> > ### Author Response · Authors · 2022-08-26
> > **continue**
> >
> > > For experiment 6.2, only qualitative results are shown. More statistics would be helpful to determining the contribution of each module compared to the baseline. For example success rate, avg. time/step taken to complete the task, etc. Without them, the experiment is not robust enough to support the claim that the method produces better latent space for planning and reasoning for robotic tasks.
> >
> > In Sec. E.5 (appendix), we present quantitative results for this experiment reporting the tree size to find a solution, including comparisons to dense GNN and control tree baselines. Other baselines were not able to complete the task at all, as the forward predictions were not stable enough to support long-horizon planning in a sparse reward setting. These results indicate that our method produces a latent space better suited for planning and reasoning within robotics tasks. We added more text at the end of Sec. E.5 in response to your comment.
> >
> > > Similarly, only qualitative results are shown for the real world experiments.
> >
> > Figures 21, 22, 23, 24, 25, 28 (appendix) show quantitative results for various real-world experiments, including baseline comparisons, generalization experiments.
> >
> > > According to the video, the image prediction is no longer sharp when the objects are textured. This makes the visual reconstruction quality questionable as the proposed method might not be able to scale to more complex objects (in terms of shape and textures).
> >
> > Our method provides significantly sharper predictions, especially sharper long term predictions, than all baselines.

---

### Official Review · Reviewer_ZKto · 2022-07-30

**Originality:** Good
**Technical Quality:** Very Good
**Clarity Of Presentation:** Very Good
**Impact:** 4

**Recommendation:**

Weak Accept: I recommend accepting the paper, but will not argue for my recommendation if the majority of other reviewers have a different opinion.

**Summary:**

The authors propose to compositionally encoder scene RGB observations into implicit neural representations. The dynamics is represented as a factored graph. The authors demonstrated the validity of their approach with various experiments.

**Issues:**

* buggy supplementary bibliography

**Quality Of The Limitations Section:**

Additional details required

**Reviewer Expertise:**

4: The reviewer is confident but not absolutely certain that the evaluation is correct

**Robotics Focus:**

Sufficient demonstration on hardware

**Strengths And Weaknesses:**

Strength:
* The paper is very well written. I had a great time going over the paper with all the interesting details. Things are presented in a clear manner, and a lot of content is packed into the short pages.
* The method is quite reasonable. It captures the "3D-ness" of objects, as well as their more lower dimensional dynamics relationship.
* A lot of the relevant works are provided, showcasing the authors' rich knowledge on the field. This is also helpful for the readers.

Weakness:
* I think some quite related recent works along the direction is not cited. For example, using implicit neural representation to represent deformable object dynamics: ACID: Action-Conditional Implicit Visual Dynamics for Deformable Object Manipulation Bokui Shen, Zhenyu Jiang, Christopher Choy, Leonidas J. Guibas, Silvio Savarese, Anima Anandkumar, Yuke Zhu, RSS, 2022.       NeRF-Supervision: Learning Dense Object Descriptors from Neural Radiance Fields. Lin Yen-Chen, Pete Florence, Jonathan T. Barron, Tsung-Yi Lin, Alberto Rodriguez, Phillip Isola. In ICRA 2022.
* I would love to hear more about the authors' thoughts on generalization. I think this might be a limitation of the approach that's worth discussing. When working with an open set of objects, will such approach still work? It will be really hard to keep a large library of INRs, and the GNN will have a difficult time reasoning if its library of latent codes grow more and more.
* Supplementary material file doesn't have a bibliography. Seems like a bug.

**Summary Of Recommendation:**

The paper is an acceptance in my opinion. However, I would love to hear more from the authors about how they see such approach generalizing and being used on more objects that are unseen.

---

> ### Author Response · Authors · 2022-08-26
> **Thank you for your review!**
>
> **Comment:**
>
> Thank you very much for your detailed and constructive review!
>
> > I think some quite related recent works along the direction is not cited [...]
>
> Thank you for pointing us to these works. We have added and discussed them in the revised version of the paper. Respective changes are colored in blue in the PDF.
>
> > I would love to hear more about the authors' thoughts on generalization. I think this might be a limitation of the approach that's worth discussing. When working with an open set of objects, will such approach still work? It will be really hard to keep a large library of INRs, and the GNN will have a difficult time reasoning if its library of latent codes grow more and more.
>
> Thank you for this constructive comment. We agree that your question deserves more discussion. Most NeRF-based methods use an unconditional network to represent a single scene (i.e., there is no generalization to different scenes/objects). In contrast, through the latent space, our approach generalizes to different object shapes, object poses in the scene, and colors within object categories similar to the training distribution. We currently use a single neural network to represent all objects; therefore, all variations in object appearance are controlled by the latent vector of size 64. Despite this being a rather small latent space, the method already exhibits interpolation capabilities for object shapes and poses. We did not have to increase the size yet to represent the variety of our scenes, including the deformable object. Therefore, we are confident that by exposing the method to a more diverse set of objects and potentially increasing the size of the latent space, one could see generalization capabilities beyond the training object category distribution, although we agree that it is an open question.
>
> We would like to mention that we are not aware of any method for visual dynamics learning where this concern of scaling to open object sets beyond a training distribution of object categories does not apply to.
>
> However, we think that we improve the state-of-the-art in this direction by introducing compositionality and, as mentioned, our method already generalizes (within the same category) over different object sizes, which is an improvement to many dynamic NeRF methods where only a single dynamic scene is interpolated in time. In addition to compositionality on a scene level, objects themselves could also be represented in a composable way, which has the potential to further increase the generalization capabilities to more diverse sets of objects.
>
> To further support this claim that we are confident of scaling the approach, our results contain an experiment where two objects with complicated shapes interact, whose interactions have never been part of the training data. Please refer to the video at timestamp 2:53 and Sec. F.1. The results show that also in this case the GNN model is able to predict the dynamics, which indicates that the model already has capabilities in generalizing over combinations of objects not seen during training.
>
> We have added a more detailed discussion about your point in the paper as well in Sec. 7.
>
> > Supplementary material file doesn't have a bibliography. Seems like a bug.
>
> Thank you. We have uploaded a revised version of the paper which includes both the appendix and the main text in one pdf, fixing this issue.
>
>
> **Zip File:**
>
> /attachment/634a4267410caf0fef06b06a9bf8cb0cb30e6415.zip

---

### Official Review · Reviewer_35gm · 2022-07-31

**Originality:** Good
**Technical Quality:** Fair
**Clarity Of Presentation:** Good
**Impact:** 3

**Recommendation:**

Weak Accept: I recommend accepting the paper, but will not argue for my recommendation if the majority of other reviewers have a different opinion.

**Summary:**

This paper studies the problem of multi-object video/motion prediction problem in robotics. Specifically, the framework consists of three modules: the encoder, GNN dynamics model, and NeRF decoder. First, the encoder takes in initial observations of the scene from multiple views and computes feature volume and latent variables for each individual object. Second, the NeRF decoder renders the view for reconstruction. Third, the GNN dynamics model learns to predict future latent variables using the adjacency matrix inferred from the NeRF density between each of the two objects. In practice, the encoder and decoder models are first trained end-to-end using the L2 image reconstruction loss, and GNN is further learned using the inferred latent variables. Experiments are mainly conducted on the non-public synthetic dataset and a few real objects (up to 4).

**Issues:**

Mentioned in [W1-W4].

**Quality Of The Limitations Section:**

Additional details required

**Reviewer Expertise:**

5: The reviewer is absolutely certain that the evaluation is correct and very familiar with the relevant literature

**Robotics Focus:**

Sufficient demonstration on hardware

**Strengths And Weaknesses:**

Strengths

* [S1] This paper studies an important task in robotic manipulation by leveraging vision signals (e.g., appearance, motion, and shape) for representation learning.
* [S2] The paper provides sufficient technical details (especially in the Supp) for replication.

Weaknesses

* [W1] The proposed framework assumes that accurate object masks are given as input to the model, which is not realistic. Although this has been discussed in section 7, the reviewer would like to know how robust the proposed model is given inaccurate masks provided using off-the-shelf instance segmentation & tracking models. Ablation studies on this are highly recommended for the next version of the paper.
* [W1.1] With inaccurate object masks, the reviewer suspects the proposed method may suffer from sudden lighting changes or ID switches (e.g., all objects with the same appearance). It would be great to clarify this.

* [W2] The motion dynamics shown in the paper and demo are easily predictable using a simple regression model. For example, if the objects of interest are articulated intelligent agents (e.g., toys) which can move by themselves, motion prediction involves a higher level of uncertainty and multi-modality. The reviewer fails to see how does GNN model handle this in the current setting. Please discuss this more explicitly as the limitation of the proposed model.

* [W3] This paper conducts experiments on non-public datasets, which makes it a bit challenging to understand the significance of the work. In addition, experimental evaluations only include ablations of different components of their proposed method rather than existing SoTA implementations. The reviewer highly recommends adding results on public benchmarks (e.g., Multi-Object Datasets and Kubrik [NewRef1-2]) even without the dynamic GNN.

* [W4] Besides prior work [31-35], there is more recent work on modeling dynamic scenes using compositional NeRFs (e.g. [NewRef3]). It would be good to discuss these existing works and consider them as potential baselines in the next version of the paper.

References
* [NewRef1] Multi-Object Datasets. [GitHub](https://github.com/deepmind/multi_object_datasets).
* [NewRef2] Kubric: A scalable dataset generator, In CVPR 2022.
* [NewRef3] Panoptic Neural Fields: A Semantic Object-Aware Neural Scene Representation, In CVPR 2022.

**Summary Of Recommendation:**

This is a borderline paper. On the positive side, it studies an important problem with meaningful extensions to the previous work. However, the reviewer holds reservations due to the strong assumption on object mask, over-simplified motion setting, and significance of the work. The reviewer would also like to exchange ideas with peer reviewers after the rebuttal.

---

> ### Author Response · Authors · 2022-08-26
> **Thank you for your review!**
>
> **Comment:**
>
> Thank you for your detailed and constructive review.
>
> > [W1] The proposed framework assumes that accurate object masks are given [...] the reviewer would like to know how robust the proposed model is given inaccurate masks provided using off-the-shelf instance segmentation & tracking models. Ablation studies on this are highly recommended for the next version of the paper.
>
> Thank you for your excellent suggestion, and we believe this is an important ablation study that we have decided to include in the current version of this paper. Following your suggestion, we have added ablation studies showing the robustness of our method to severe mask perturbations, presented in Sec. E.4, Fig. 16, Fig. 17, Sec. F.2, and Fig. 26 as well as the video attached to this comment. Results indicate that our method does not rely on precise masks. Furthermore, we have applied the method to real-world scenarios where we rely on off-the-shelf segmentation methods, which do not provide accurate masks, further supporting our claim that precise masks are not necessary.
>
> > [W1.1] With inaccurate object masks, the reviewer suspects the proposed method may suffer from sudden lighting changes or ID switches (e.g., all objects with the same appearance). It would be great to clarify this.
>
> As mentioned above, the method is robust against inaccurate object masks. Due to the fact that we train the GNN dynamics model separately from the encoder/decoder (as this was sufficient), during training of the GNN, the method requires correspondences of object identities between two consecutive time steps, which was not an issue to achieve in our scenario as the objects were sufficiently distinct from each other.
> However, at inference time, masks are required only for the initial scene, i.e., no tracking of object identities is required during prediction. In particular, the object encoder and dynamics model are permutation invariant. Hence, changing object IDs will not change the behavior of the model during prediction.
>
> > [W2] The motion dynamics shown in the paper and demo are easily predictable using a simple regression model.
>
> Pushing dynamics of multiple interacting objects is considered a very challenging prediction task [66, 67, 68, 69]. Predicting multi-object interactions requires the model to reason about contact states which rely on a precise understanding of geometry, friction parameters, center-of-masses, and inertia parameters, all of which our method learns to extract implicitly from videos showing interactions of the objects. Generalizing to different numbers of objects in scenes where multiple objects can interact within one timestep or even deformable objects further complicates the task.
>
> >For example, if the objects of interest are articulated intelligent agents (e.g., toys) which can move by themselves, motion prediction involves a higher level of uncertainty and multi-modality. The reviewer fails to see how does GNN model handle this in the current setting. Please discuss this more explicitly as the limitation of the proposed model.
>
> We would kindly like to ask you to be more elaborative about “articulated intelligent agents”.
> If articulated objects are meant in the sense of degrees of freedom that are subject to constraints like joints, then we have included a video in response to your comment showing that our framework can be applied to an articulated object, a sliding door, which is subject to motion constraints. Also, in this case, the GNN can predict the dynamics of the scene well.
>
> If, on the other hand, multi-agent scenarios are meant where certain objects behave autonomously without being driven by controlled actions, then this is not explicitly being taken care of in our current formulation. However, there is nothing inherent in our architecture that would prevent applying the method to such scenarios. For example, one could add memory in terms of recurrent mechanisms to the GNN in order to learn the behavior of the autonomous agent. We agree that in this case there is a much higher uncertainty in the system, but it is not the main problem we investigate in this work, and hence leave such cases as future explorations.
> We are happy to discuss this further in the paper if you clarify what exactly you are referring to.
>
>
> **Zip File:**
>
> /attachment/3d24e44900e1b31216b88145a087ba28d526f81e.zip

---

> > ### Author Response · Authors · 2022-08-26
> > **continue**
> >
> > > [W3] This paper conducts experiments on non-public datasets, which makes it a bit challenging to understand the significance of the work.
> >
> > Thank you for raising this point. We will release our datasets to the public upon the acceptance of the paper. The reason for choosing our own environments is that we want to study not only static scenes, but also dynamic ones where we additionally have the possibility to apply actions to objects and define loss functions to test our planning algorithm. We are not aware of a public dataset for this multi-object, multi-view setup that allows us to apply actions and define loss functions.
> >
> > > The reviewer highly recommends adding results on public benchmarks (e.g., Multi-Object Datasets and Kubrik [NewRef1-2]) even without the dynamic GNN.
> >
> > Thank you for pointing this out. Following your suggestion, we have applied our method to the CLEVR 3D dataset from [38]. For results, see the paper Sec. E.6.
> >
> > > In addition, experimental evaluations only include ablations of different components of their proposed method rather than existing SoTA implementations.
> >
> > Our baseline comparisons include two SoTA implementations, the global NeRF dynamics model learning and the CNN-based baseline, both from [2].
> >
> > > [W4] Besides prior work [31-35], there is more recent work on modeling dynamic scenes using compositional NeRFs (e.g. [NewRef3]).
> >
> > Thank you for pointing us to this work. We have added and discussed it in the revised version of the paper.

---

> > ### Author Response · Authors · 2022-08-26
> > **Thank you for your reply!**
> >
> > We are adding our reply to your reply here again, as openreview did not allow us to select you as a reader.
> >
> > > * Regarding [W1] & [W1.1]: Thanks for ablation studies! The experimental results partially addressed my concerns in the initial review. However, I can still see weird floaters (in black) from the uploaded videos due to noise pattern injected or imperfect mask.
> > > * Regarding [W2]: Thanks for uploading the sliding door video! I was referring “articulated intelligent agents" to (toy) animals (e.g., cat, dog) that can move by themselves.
> > > * Regarding [W3]: Thanks for adding CLEVR 3D dataset in the paper!
> > > * Regarding [W4]: The discussion looks good.
> >
> >
> > Regarding [W1] & [W1.1]: We agree that there are small artifacts from imprecise masks. However, given the amount of mask perturbation, we would argue that the method does a very good job in dealing with these imprecise masks. For example, the masks for the blue pusher in the video are severely perturbed given their size (approx. 40% in some of the views). Also, for the purple object, a significant chuck is missing in the masks. We have no indications that these artifacts did have a significant impact in learning the dynamics models on top of the learned latent space, as also shown by the ablation study in the updated paper.
> >
> > Regarding [W2]: Thank you for the clarification. We have added the following discussion regarding this point in the PDF (attached) in Sec. 7 at the bottom:
> >
> > Autonomous Dynamical Systems: The types of dynamical systems we considered in this work are such that if no actions are applied to objects in the scene, they remain in their motion state (i.e., the objects do not have any internal actuators). Therefore, if the scene contains objects that exhibit autonomous behavior, then the current problem formulation cannot predict the dynamics of these autonomous objects. One way of addressing this is to introduce memory into the dynamics model (e.g., in terms of recurrent mechanisms) or to build agent models (e.g., through imitation learning) to learn the behavior of autonomously behaving objects on top of the learned latent space. We leave the exploration in this direction for future work.
> >
> > ---
> >
> > Thank you again for the constructive feedback, which allowed us to greatly strengthen the paper with additional results and more well-rounded discussions. Please let us know if you have any additional concerns, and we kindly request if there is a chance for you to increase your score.

---

### Official Review · Reviewer_d3gL · 2022-08-06

**Originality:** Very Good
**Technical Quality:** Very Good
**Clarity Of Presentation:** Very Good
**Impact:** 3

**Recommendation:**

Weak Reject: I recommend rejecting the paper, but will not argue for my recommendation if the majority of other reviewers have a different opinion.

**Summary:**

The authors propose a method for learning dynamics from multi-view series data of an environment consisting of multiple objects. The proposed method has NeRF to include 3D inductive bias and GNN to extract the compositional representation to enable long-term prediction. Experiments on both simulators and real-world situations confirm the effectiveness of the proposed method.

**Issues:**

I would like you to work on the concerns I have listed in "weaknesses". If you respond appropriately to these and revise your paper to reflect them, I will consider raising the score.

**Quality Of The Limitations Section:**

Additional details required

**Reviewer Expertise:**

3: The reviewer is fairly confident that the evaluation is correct

**Robotics Focus:**

Sufficient demonstration on hardware

**Strengths And Weaknesses:**

Strengths:
- The paper is well organized and easy to read.
- The proposed method is valid and novel compared to existing NeRFs.
- Experiments confirm its effectiveness on complex datasets of multiple objects. In particular, it is very significant that it is demonstrated to work properly on deformable objects and objects in real-world environments.

Weaknesses:
- In this paper, the authors aim to obtain object-specific representations from multi-view images and to predict future observations. However, the authors do not mention many related works on similar tasks (although some are cited, such as [32]). For example, many studies on acquiring object-specific representations from images or videos for segmentation and prediction have been studied as object-centric representation learning (or object-centric world model) (1,2,3,4,5). In particular, in (5), object-specific representations are acquired from videos using graph neural networks. There is also work on learning object-centric representations in multi-view (6). I think this proposed method has novelty compared to these in that it acquires object representations in multi-view and time series, but the authors should cite these studies and explicitly discuss the differences.
(1) MONet: Unsupervised Scene Decomposition and Representation
(2) Multi-Object Representation Learning with Iterative Variational Inference
(3) Entity Abstraction in Visual Model-Based Reinforcement Learning
(4) Improving Generative Imagination in Object-Centric World Models
(5) Contrastive Learning of Structured World Models
(6) Learning Object-Centric Representations of Multi-Object Scenes from Multiple Views
- The authors state that "we believe having masks is a reasonable assumption," but many of the above object-centric methods do not assume that masks are given, and acquire object-specific representations only from images and videos. Therefore, it is not obvious that this assumption is reasonable in my opinion. The authors would like more explanation as to what perspective this assumption is valid compared to related studies.

**Summary Of Recommendation:**

The method is novel and its ability to make long-term predictions in the real world is a significant contribution; however, as mentioned above, I still have concerns regarding the lack of citation of previous studies and the appropriateness of the problem setting. Therefore, I consider it a weak rejection.

---

> ### Author Response · Authors · 2022-08-26
> **Thank you for your review!**
>
> **Comment:**
>
> Thank you for your detailed and constructive review. We are happy that you acknowledge that our method providing “long-term predictions in the real world is a significant contribution”. Our answers to your concerns about missing related work and the mask assumption are below.
>
> > However, the authors do not mention many related works on similar tasks
>
> Thank you for pointing us to these works; we have added and discussed them in the revised version. The newly added text is colored in blue.
>
> > [...] masks are given [...] more explanation as to what perspective this assumption is valid compared to related studies
>
> Thank you for this comment; we agree that the masks require more discussion and attention.
> In many of the related studies you mentioned, the focus is on unsupervised object discovery.
> This focus is in some way independent from our work, where we focus on proposing a scene representation framework from images that supports learning dynamics models while achieving significantly stronger long-horizon prediction stability by incorporating 3D biases such that they become suitable for planning. We believe that the techniques developed in these studies you mentioned could be integrated into our framework to discover the objects in an unsupervised way as well.
>
> Following your comments, we have added a more detailed discussion in the paper about the mask assumption (Sec. 7).
>
> Additionally, we have added several experiments (cf. Sec. E.4, Fig. 16, Fig. 17, Sec. F.2, and Fig. 26 as well as the video attached to this comment) showing the robustness of our method to severe mask perturbations. Results indicate that precise masks are not required for our method to work successfully. Further, we have shown that our method can work in the real world without access to ground-truth masks. Extracting masks for the real-robot experiments was not a major concern. Generally, we don’t think that object masks are a strong assumption, as our method is not only robust against imprecise masks as shown in the new experiments, but there is also significant progress in instance segmentation methods in the computer vision community — both supervised and unsupervised as you have suggested. In relation to related work that discovers objects without masks, we believe that techniques such as slot-attention used in [38, 77] could be employed in our method as well. Since unsupervised instance segmentation is not the main focus of this paper, we leave the exploration of integrating such modules in our pipeline for future works.
>
> We would like to add other advantages our method provides by including masks as input. Most approaches for learning actionable dynamics models assume that actions are direct inputs to the model. Therefore, they typically overfit to this action input (e.g., in terms of movements applied to specific objects). The dataset to train such dynamics models, therefore, also has to contain the action, which is questionable if that is available in all circumstances.
> In contrast, our approach does not rely on a dataset that contains specific actions for the dynamics model. For our method, it is sufficient that the dataset contains videos of objects interacting with each other. To train the dynamics model, one just requires to know which object was the actuated one (e.g., in terms of the mask of the pusher in our experiments) and no extracted action. It would also be possible to have different objects in different examples being used as the pusher. Although knowledge of this is also additional information, we believe that extracting which object was the actuated one is much easier than additionally extracting the movement of that object in terms of an action. At inference time, we can also use the dynamics model not only with the object being used as a pusher as in the training set, but any arbitrary object and the model can handle this directly.
>
> Moreover, with the masks, we can also identify objects in order to define loss functions for a downstream planning task flexibly, which is not directly possible for unsupervised object discovery methods.
>
> To summarize, having access to masks was not a strong assumption for our method, as they don’t have to be accurate for our method to be successful. Future work could investigate integrating unsupervised object discovery methods into our framework.
>
>
> **Zip File:**
>
> /attachment/c187163d8bc68689f89d1b849f65d48354705230.zip

---

### Author Response · Authors · 2022-08-26
**Response by the Authors**

**Comment:**

We thank the reviewers for their detailed reviews.
Reviewers agree that the work is novel (“method is valid and novel compared to existing NeRFs”, ”The work is novel”, “significant contribution”, “important task in robotic manipulation”), and acknowledge the experiments (“Experiments confirm its effectiveness on complex datasets of multiple objects.”, “very significant that it is demonstrated to work properly on deformable objects and objects in real-world environments”), experimental evaluation (“Comprehensive ablation studies are performed to support the design choices of the model components”, “The authors demonstrated the validity of their approach with various experiments.”) and generality of the approach (“generalized to the number of objects and handles the interaction between objects well, even when the object is deformable”, “ works on non-trivial deformable objects”).
Further, the reviewers acknowledged the writing of the paper (“The paper is very well written. I had a great time going over the paper with all the interesting details. Things are presented in a clear manner”, “The paper is well-organised and easy to read”).

Based on the reviewers’ comments, we made the following changes to the paper:

* Mask perturbation experiment: The main critique raised by reviewers d3gL and 35gm is that our method assumes object masks as input. We have added several experiments (quantitative and qualitative) showing that precise masks are not necessary for our method to be successful. Further, we discuss this assumption in more detail below and in the updated version of the paper.
* Related work: We have added and discussed all related work mentioned by the reviewers d3gL, 35gm, and ZKto
* Experiments on the CLEVR 3D dataset (Rev. 35gm)
* Several clarifications and additions to the paper in response to Rev. eGyX
* More detailed discussion related to scaling to open object sets (Rev. ZKto)

Respective changes in the PDF are marked in blue.


**Zip File:**

/attachment/5be532e245a064a7cb15c4f1f2eb3de1a7e09a50.zip

---

### Meta-Review · Area_Chair_ifR4 · 2022-08-08

**Recommendation:** Accept (Poster)
**Confidence:** 4

**Metareview:**

This paper introduces a dynamics model for multi objects that combines a GNN and Neural radiance field. The paper shows that the improved rendering this allows for longer horizon dynamics prediction as it results in an improved latent space, and that compositional dynamics is important for dynamics prediction. Results are impressive, in both simulation and real-world scenes, with MPC used to demonstrate control using the learned vision-based dynamics model (although it would be interesting to see this scale to scenes with more complex backgrounds). The original review process highlighted that:

* The paper is well-organised and easy to read
* The work is novel, and results show that it works on non-trivial deformable objects
* The NeRF aids in the learning of 3D inductive biases, and important property lacking in more free-form approaches

but reviewers expressed:
* Some concerns about generality, scaling to open set objects
* Concerns about a requirement that objects masks are already known
* and queried some missing related work

The authors provided a strong rebuttal to these reviews (some scalability already present, additional experiments around the effects of object mask uncertainty, additional discussion of related work). In light of this, I recommend acceptance.

---

> ### Author Response · Authors · 2022-08-26
> **Thank you for your review!**
>
> **Comment:**
>
> > Concerns about a requirement that objects masks are already known
>
> Based on the reviewers’ suggestions, we have added several experiments showing the robustness of our method to mask perturbations. The results indicate that precise masks are not required for our method to work successfully. Further, we have shown that our method works in the real world where no precise masks are available, and extracting them was not a major concern in our experiments. Generally, we don’t think that object masks are a strong assumption, as our method is not only robust against imprecise masks as shown in the new experiments and the real-world experiments which already had imprecise masks, but there is also significant progress in instance segmentation methods in the computer vision community - both supervised and unsupervised as has also been suggested by the reviewers. As discussed below in the comments to the reviewers, having masks as input has other advantages, namely that, when learning actionable dynamics models, our method requires fewer assumptions and is more flexible.
>
> A more detailed discussion about the mask assumption, including the new experiments, was added to the paper (Sec. 7, Sec. E.4, Fig. 16, Fig. 17, Sec. F.2, and Fig. 26).
>
> > Some concerns about generality, scaling to open set objects
>
> Thank you, this is an interesting point that deserves more discussion. Most NeRF-based methods use an unconditional network to represent a single scene (i.e., there is no generalization to different scenes/objects). In contrast, through the latent space, our approach generalizes to different object shapes, object poses in the scene, and colors for object categories within the training distribution. We currently use a single neural network to represent all objects; therefore, all variations in object appearance are controlled by the latent vector of size 64. Despite this being a rather small latent space, the method already exhibits interpolation capabilities for object shapes and poses. We did not have to increase the size yet to represent the variety of our scenes, including the deformable object. Therefore, we are confident that by exposing the method to a more diverse set of objects and potentially increasing the size of the latent space, one could see generalization capabilities beyond the training object category distribution, although we agree that it is an open question.
>
> We would like to mention that we are not aware of any method for visual dynamics learning where this concern of scaling to open object sets beyond a training distribution of object categories does not apply to.
>
> However, we think that we improve the state-of-the-art in this direction by introducing compositionality and, as mentioned, our method already generalizes (within the same category) over different object sizes and poses, which is an improvement to many dynamic NeRF methods where only a single dynamic scene is interpolated in time.
>
> To further support this claim that we are confident of scaling the approach, our results contain an experiment where two objects with complicated shapes interact, whose interactions have never been part of the training data. Please refer to the main video at timestamp 2:53 and Sec. F.1, and Fig. 25. The results show that also in this case the GNN model is able to predict the dynamics, which indicates that the model already has capabilities in generalizing over combinations of objects not seen during training.
>
> > Some missing related work
>
> We thank the reviewers for pointing us to more related work. We have added and discussed all related work suggested by the reviewers, marked in blue in the paper.
>
>
> **Zip File:**
>
> /attachment/f7a43336f3caa83a5b6b6bf129da9b2e5e1ae334.zip